# FORMALIZING AUDITS OF ML MODELS AS A SEQUENTIAL DECISION-MAKING PROBLEM

## ABSTRACT

Auditing is a governance mechanism for evaluating ML models to identify and mitigate potential risks. This process is critical, as undetected issues in models, such as incorrect predictions or inappropriate feature use, can lead to adverse consequences. In this work, we focus on application audits, which aim to detect errors in domain-specific ML applications. Application audits are important as they assess the risks posed by ML models to guide mitigation. Currently, application audits are predominantly manual, relying on domain experts to identify model errors by inspecting predictions and their explanations, which limits the scalability of audits. To complement human auditors, we explore algorithmic approaches to application auditing and formalize the auditing task as a sequential decision-making problem. We propose SAFAAI, a novel conceptual framework for auditing, inspired by principles of situational awareness, to formally define the objectives of application audit problem. Building on this foundation, we introduce RLAuditor, a reinforcement learning method for automating application audits of ML models. We validate our approach on multiple ML models and datasets, both with and without human auditors, demonstrating its effectiveness in facilitating audits across different contexts. To our knowledge, this work is the first to formalize application audits for ML models as a sequential decision-making problem, informing the design of future automated and human-AI collaborative auditing approaches.

## 1 INTRODUCTION

Auditing is a governance mechanism that evaluates machine learning (ML) models by analyzing their outcomes or simulating user interactions to identify, assess, and mitigate potential risks and harms (Mökander et al., 2023; Lam et al., 2024; Raji & Buolamwini, 2019). Auditing ML models is crucial for ensuring their reliability and safety, particularly in high-stakes applications (Kelly et al., 2019; Zhang et al., 2020). For instance, in embodied AI systems like autonomous vehicles, perceptual errors caused by ML models can lead to accidents and, in severe cases, loss of life (Cummings, 2021; 2023; Cummings & Bauchwitz, 2024). In healthcare, errors in ML-based diagnostic models can lead to misdiagnoses or inappropriate treatments, potentially endangering patient safety (Sheliemina et al., 2024; Yu et al., 2024). Regulatory frameworks such as the European AI Act mandate audits and testing to identify vulnerabilities, biases, and unintended behaviors in ML models. To formalize the concept of ML model auditing, Mökander et al. (2023) propose a three-layered framework consisting of governance audits, model audits, and application audits. While the first two types are designed to be conducted by model technology providers, i.e. the organizations developing ML models, the third focuses on downstream applications and must be conducted within domain-specific contexts.

In this paper, we focus on this third type, known as **application audits**. Adapting the taxonomy of Mökander et al. (2023) for general ML models, application audits can be defined as: "impact-oriented assessment of the risks posed by products and services built on top of pre-trained ML models." Application auditing *need to be conducted by end-users or application developers* who rely on end-user ML tools to build and evaluate customized applications by fine-tuning general-purpose ML models. Unlike large organizations that develop the pre-trained ML model, these users may have limited computational resources and may lack dedicated teams for model auditing. Moreover, *application audits must be conducted routinely* to address evolving requirements within specific domains.

**End-users currently rely on manual processes to conduct application audits.** Figure 1 illustrates a typical approach to application auditing. A human expert queries the model using an auditing dataset, observing its behavior, including *decisions and explanations* for the test samples. Alongside data samples, model explanations aid the audit process by revealing feature use and supporting the assessment of model behavior (Yadav et al.; Balayn et al., 2022); their usefulness is also validated in our human user study. The

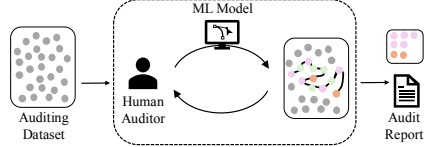

Figure 1: Application audits help identify problematic model behaviors in domain-specific ML applications.

auditors identify erroneous samples and then iteratively select additional test samples from the dataset to further probe the model. The selection of test samples is guided by intuition to maximize the detection of problematic model behaviors. Identified behaviors are then categorized based on factors such as severity and error type, culminating in an audit report.[1] The entirely manual nature of the auditing process limits scalability, hinders the discovery of problematic model behaviors, and leads to inefficient use of domain experts' time. **Therefore, more research is needed on automated auditing methods that complement existing manual approaches** to better utilize domain experts' time and identify problematic model behaviors. Ideally, such methods should provide a general-purpose solution that can be adapted to domain-specific applications.

To achieve this, our **first contribution is the formalization of selecting informative test samples as a sequential decision-making problem**. Our insight is that this process is inherently sequential in nature. Auditors typically start by evaluating model predictions and explanations on a few exploratory instances to understand the model's performance and reasoning. They then use this knowledge to strategically select subsequent instances to reveal informative behavior. As familiarity grows, auditors become more effective at identifying errors, continuously balancing exploration (learning the model) and exploitation (using that knowledge to detect mistakes).

Our **second contribution is to address the gap in automating the auditing process by proposing a structured conceptual framework, SAFAAI**. It formally defines urgent, underexplored research challenges in application audits by introducing three levels of audit goals, analogous to the three levels of situational awareness (Endsley, 1995; Sanneman & Shah, 2022). These levels help define concrete optimization objectives for mathematical modeling of the application auditing problem.

Building on the first two contributions, our **third contribution is the use of reinforcement learning (RL) as an algorithmic solution to support auditing.** To mathematically formalize this sequential process, we model the auditing process as a Markov decision process (MDP) by designing the state (knowledge about the model) and the action (selection of subsequent test samples). We explore various choices of MDP state representations, including those involving model explanations, following the strategies used by human experts (Yadav et al.; Balayn et al., 2022). Our sequential decision-making model, combined with an off-the-shelf RL algorithm, culminates in RLAuditor: an auditing agent to select test samples (actions) based on the current knowledge of the model (state).

In both numerical and human subject experiments, we observe that RLAuditor enables efficient discovery of model errors, while effectively balancing the trade-off between exploration and exploitation. Figure 2 illustrates this exploration–exploitation process during auditing, showing training curves for RLAuditor in a domain-specific context. Similar to human auditors, the RL agent first explores the model behavior on test samples to learn a better state representation. As the state representation becomes more accurate, the RL agent then exploits its knowledge of the model error to identify an increasing number of model errors. Through experiments on multiple ML models, including one trained on medical imaging dataset, we demonstrate the effectiveness of this RL-based approach in facilitating both automated and human-AI collaborative auditing of ML models.

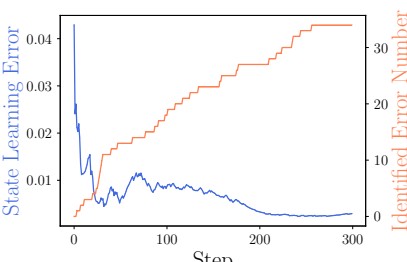

Figure 2: Application auditing is a procedure for acquiring accurate knowledge of the model (blue) to identify errors (orange).

---

[1] A practical instance of this manual auditing process is detailed in Section 5.

## 2 RELATED WORK

Research on auditing ML models has grown in response to the need for responsible AI deployment (Mökander et al., 2023; 2022). Existing methods can be grouped into two categories: facilitating human audits and automating the audit process. Model explanations play a key role in both, revealing feature use and supporting assessment of model behavior (Yadav et al.; Balayn et al., 2022).

**Facilitating Manual Audits.** Tools in this category often provide application-specific interfaces for exploring model behaviors (Balayn et al., 2022; Wexler et al., 2019; Ren et al., 2016; Zhang et al., 2018). For example, the What-If Tool (Wexler et al., 2019) is a model-agnostic interactive visualization tool for analyzing ML models. Another approach leverages generative models to expand auditing datasets, with humans reviewing and curating the most relevant test cases (Ribeiro & Lundberg, 2022; Rastogi et al., 2023; van Breugel et al., 2024). While useful, these approaches rely on human effort, limiting scalability.

**Automated Audits.** Our work aligns with the automated methods which aim to detect undesirable behaviors without heavy human involvement (Kang et al., 2018; Ma et al., 2018; Singla et al., 2021; Lourenço et al., 2019). Examples include assertion-based detection for specific tasks (Kang et al., 2018) or probabilistic models for ranking potentially erroneous labels in autonomous vehicle datasets (Kang et al., 2022). Existing approaches are largely domain-specific, whereas our method targets general applicability across diverse models and datasets.

**Role of Model Explanations.** Explanations support audits in both manual and automated settings. User studies demonstrate their effectiveness in manual audits (Balayn et al., 2022; Yadav et al.), while automated approaches such as XAudit (Yadav et al.) uses explanations to reconstruct models and ensure the inclusion of relevant features. However, XAudit is limited to simple models, such as linear regression. In contrast, our work utilizes explanations to audit more complex models, including CNNs for image classification. A more comprehensive review of related work is in Appendix A.

## 3 FORMALIZATION AND ALGORITHMIC SOLUTION

This section formalizes the auditing task as a sequential decision-making problem and presents our algorithmic solution. First, we introduce SAFAAI, a novel conceptual framework that formally defines the objectives of the application audit problem. Building on this foundation, we design an MDP model to develop RLAuditor: an RL agent that automates the selection of test samples during application audits of ML models.

### 3.1 SAFAAI: SITUATIONAL AWARENESS FRAMEWORK FOR APPLICATION AUDITS OF AI

Situational Awareness (SA) refers to the comprehension of environmental conditions, including relevant system parameters, and has been extensively studied in human factors literature, particularly in the context of human-automation teams working in complex environments (Endsley, 1988).

Observing that application audits require the auditor to have situational awareness of the model (the system) within its application context (the environment reflecting deployment on an application-specific dataset), we build on the general SA framework to introduce the Situational Awareness Framework for Application Audits of AI (SAFAAI).[2]

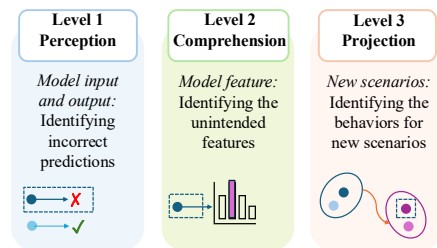

**Definition 3.1** (Situational Awareness Framework for Application Audits of AI). In the context of application auditing, situational awareness of an ML model $\mathcal{M} : X \rightarrow Y$ is defined as the detection of ML model $\mathcal{M}$ errors in the auditing dataset $\mathcal{D}$, the comprehension of incorrect reasoning processes of the ML model, and

Figure 3: Illustration of SAFAAI.

the projection of model errors in future application-specific contexts that may not be present in the auditing dataset. SAFAAI includes three levels of situational awareness, as shown in Figure 3.

---

[2]Coincidentally, SAFAAI means *cleanliness* or *clarity* in Hindi. An apt reflection of our framework's goal: to provide auditors with clarity about an ML model's behavior in its application-specific context.

1. **Perception (Level 1)**: Identification of any incorrect model predictions in the auditing dataset. Mathematically, this translates to the detection of the subset $C \subset \mathcal{D}$ of input instances $x$ in the dataset $\mathcal{D}$ for which the model's prediction $\mathcal{M}(x) \neq y$ differ from the ground truth $y \in Y$.

2. **Comprehension (Level 2)**: Comprehension of any incorrect reasoning processes of the model. Level 2 errors target potential feature misuse, such as prohibited features or invalid combinations. These may not always cause incorrect predictions (Level 1), but must still be detected due to their potential harm. Mathematically, given a feature set $\mathcal{F}$, we formulate this as an auditor comprehending the features $F'_y \subseteq \mathcal{F}$ used by the model to predict output $y$, and assessing whether they align with the auditor-intended features $F_y \subseteq \mathcal{F}$.

3. **Projection (Level 3)**: Detection of any incorrect model predictions in the set of all application-specific model inputs $X$, including novel tasks that are not present in the auditing dataset. Analogous to the original SA framework, this third level requires the auditor to *project* model behavior onto new tasks (unseen scenarios).

Given the potentially vast input space $X$ and, consequently, the auditing dataset $\mathcal{D}$, manually identifying every instance of unintended model behavior is often impractical and resource-intensive in real-world applications. This requirement for an actionable and practical auditing framework leads us to define and address the *Efficient Application Auditing Problem*, which aims to identify the maximum number of relevant examples of model errors given limited auditing resources. Since our work is the first attempt at formalizing application auditing, we will only focus on the first two levels in the following sections and leave the exploration of the third level for future research.

**Definition 3.2** (Efficient Application Auditing Problem)**.** *Efficient* application auditing is defined as the problem of selecting $K$ input instances, denoted by the set $C'$, that maximally overlap with the set of inputs on which the model makes errors. Formally, this translates to selecting $C' \subseteq \mathcal{D}$ such that $|C'| = K$ and $|C' \cap C|$ is maximized.

### 3.2 RLAUDITOR: APPLICATION AUDITING USING REINFORCEMENT LEARNING

We now introduce **RLAuditor**, an agent for solving the *Efficient Application Auditing* problem. First, inspired by how humans audit ML models, we frame auditing as a *sequential decision-making problem* and model it as an MDP. Next, we solve the designed MDP using RL to obtain RLAuditor's auditing policy. A detailed explanation is provided in Appendix B.

**Auditing as a Markov Decision Process.** Given a black-box ML model $\mathcal{M} : X \to Y$ and auditing dataset $\mathcal{D}$, the goal of the Efficient Application Auditing problem is to select a set $C' \subseteq \mathcal{D}$ of model input instances such that $|C'| = K$ and $|C' \cap C|$ is maximized. Furthermore, based on the observation that human auditors often rely on model explanations to understand model reasoning, we assume access to model explanations for a given input-output pair in the form of *feature attribution* scores.

Formally, consider the model $\mathcal{M}$ (accompanied by an explanation algorithm) that maps an input data point $x \in X$ to a predicted class label $\hat{y} \in Y$ and an explanation $e \in \mathcal{E}$, where $e$ is a feature attribution vector. For simplicity, we overload $\hat{y}$ to refer to the full logits of the model's output. The sequential decision-making problem then translates to learning a policy $\pi$ for selecting the $(m+1)$-th element $x_{m+1}$ of the set $C'$, given the model behavior on the elements selected thus far $(x_1, \ldots, x_m)$.

To enable tractable policy computation using reinforcement learning, we approximate the policy as a Markovian policy $\pi(\cdot \mid s)$, where $s$ denotes a suitable Markovian statistic derived from the problem inputs and the previously selected elements $(x_1, \ldots, x_m)$ of $C$. Correspondingly, we frame the sequential decision-making process as an MDP. Learning effective policies in this setting hinges on the design of the state representation $s \in \mathcal{S}$, followed by the appropriate specification of the remaining elements of the MDP tuple $(\mathcal{S}, \mathcal{A}, T, R)$. In the remainder of this section, we describe the design choices for each component of the MDP, along with a reinforcement learning-based approach to compute its policy.

**State Space** $s \in \mathcal{S}$**.** To ensure effective auditing, the state $s$ should represent the auditor's knowledge of the model, based on its performance and explanations on previously selected test cases $(x_1, \ldots, x_m)$. At the same time, to enable tractable policy learning, the state $s$ should be both compact and Markovian. Designing such a state is non-trivial. In this work, we explore several potential state features and evaluate their relative advantages through ablation studies. Based on the

results of these studies (presented in Section 4.2), we adopt a state representation for RLAuditor that includes a summary of: (i) the model's prediction performance across classes, and (ii) the features used by the model, which can be reflected by feature attribution scores. Concretely, we selected $s = [\phi_{\text{class}}, \phi_{\text{explanation}}]$, where $\phi_{\text{class}} \in \mathbf{R}^{N \times N}$ is the matrix for class prediction information, $N$ is the number of classes; and $\phi_{\text{explanation}} \in \mathbf{R}^{N \times d}$ represents the transform matrix from features and classes, and $d$ is the dimension of the feature embeddings. Concretely, for $\phi_{\text{class}}$ we use the prediction logits for each viewed sample as the knowledge representation for the classifier. Each row of the matrix $\phi_{\text{class}} \in \mathbf{R}^{N \times N}$ corresponds to a predicted class, and the entry at position $(i, j)$, denoted as $\phi_{\text{class}}(i, j)$, represents the averaged logits of the classifier for class $j$ among the samples predicted as class $i$. Formally, $\phi_{\text{class}}(i, j)$ is defined as:

$$\phi_{\text{class}}(i, j) = \frac{1}{|\mathcal{U}_i|} \sum_{x_k \in \mathcal{U}_i} \hat{y}_j^i,$$

where $\mathcal{U}_i$ is the set of all *past* samples predicted as class $i$, and $\hat{y}_j^i$ is the logit output of the classifier for class $j$ given the input sample $x_k$. This representation captures the confidence distribution of the classifier across all classes for samples predicted to belong to a specific class. Similarly, we define the matrix $\phi_{\text{explanation}} \in \mathbf{R}^{N \times d}$, where each row corresponds to a specific predicted class, and each entry captures the average feature embeddings for that class. Specifically, the entry at position $(i, k)$, denoted as $\phi_{\text{explanation}}(i, k)$, represents the average embedding of feature $k$ among all samples predicted as class $i$. Formally, we define:

$$\phi_{\text{explanation}}(i, k) = \frac{1}{|\mathcal{U}_i|} \sum_{x_j \in \mathcal{U}_i} \psi_k(x_j, e_j),$$

where $\psi_k(x_j, e_j)$ denotes the feature embedding of dimension $k$ for the sample $x_j$ given $e_j$. To construct $\psi_k(x_j, e_j)$, we utilize the $e_j$ as a mask over the input data and extract the corresponding feature embeddings. These embeddings are then averaged to construct $\phi_{\text{explanation}}$, allowing us to capture class-specific feature representations.

**Action** $a \in \mathcal{A}$. The action $a = x$ simply corresponds to selecting the next element $x = x_{m+1}$ of $C'$. Thus, the Markovian policy $\pi(a \mid s)$ aims to learn how to select the next element of $C'$ based on the current information about class predictions and explanations encoded in $s$.

**Transition Model** $T(s'|s, a)$. The transition function defines how the state is updated based on the current state and the selected action. To compute the next state, we first query the model $\mathcal{M}$ on the most recently selected input $x = a$, obtaining the corresponding model prediction $\hat{y}$ and explanation $e$. Next, given the tuple $[x, \hat{y}, e]$, we define the transition for each state feature in the updated state $s' = [\phi'_{\text{class}}, \phi'_{\text{explanation}}]$ via the following deterministic functions:

$$\phi'_{\text{class}}(i) = \frac{1}{|\mathcal{U}_i| + 1} \left( |\mathcal{U}_i| \cdot \phi_{\text{class}}(i) + \hat{y} \right),$$

$$\phi'_{\text{explanation}}(i) = \frac{1}{|\mathcal{U}_i| + 1} \left( |\mathcal{U}_i| \cdot \phi_{\text{explanation}}(i) + \psi(x, e) \right).$$

**Reward Model** $R(s, a)$. To leverage RL, a well-designed reward function is essential. Following the guideline that the "reward signal is your way of communicating to the agent what you want achieved, not how you want it achieved" (Sutton, 2018), we design a reward function that encourages the algorithm to select inputs where the model exhibits errors. Intuitively, the reward function informs the auditor whether the selected action successfully leads to the discovery of model errors. Mathematically, the reward function assigns a score of 1 to selected model inputs that reveal errors (as defined by SAFAAI), and a score of 0 to those that do not. During training, assessing whether the model makes an error requires human in the loop to judge model outputs and provide this reward signal. These annotations, or human inputs, are required only during the training phase. During deployment of RLAuditor, no annotations are needed, enabling a fully automated auditing process.

**Auditing Policy** $\pi$. Lastly, given the MDP model of the auditing process $(\mathcal{S}, \mathcal{A}, T, R)$, we employ a widely used RL algorithm, Deep Q-Network (DQN), to learn the auditing policy for sequentially selecting the elements of $C'$ (Mnih et al., 2015). To learn this policy, we provide an auditing dataset denoted as $\mathcal{D}_{\text{train}} \subseteq \mathcal{X}$. We evaluate its performance on a test set $\mathcal{D}_{\text{test}} \subseteq \mathcal{X}$. Further implementation details for the training paradigm of RLAuditor are provided in Appendix B.

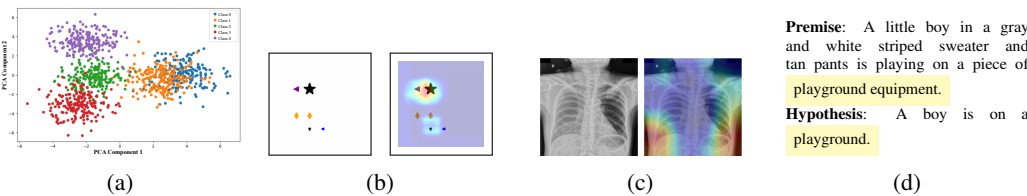

Figure 4: Example of used datasets. (a) `SynTab`: PCA analysis of class features. (b) `SynImg`, (c) VinDr-CXR and (d) e-SNIL: Input $x$ and its $e$ (highlighted in colors).

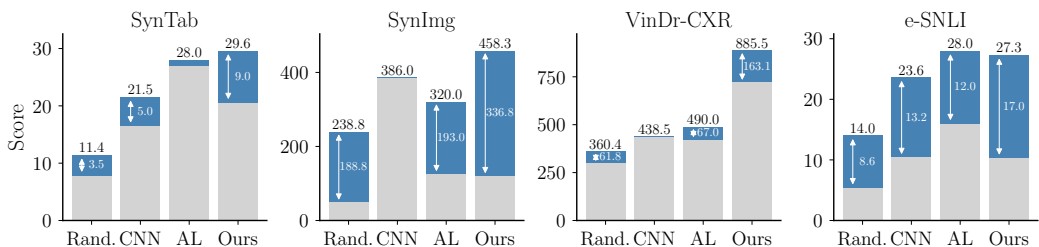

Figure 5: Auditing results for two levels of errors on fours datasets. Bar ▉ represents the number of identified Level 2 errors. The total number of identified errors in both levels is listed above bars.

## 4 EXPERIMENTS

We now compare our RLAuditor's performance against baselines in discovering Level 1 and Level 2 errors defined in SAFAAI (Section 4.1). We use four datasets for evaluation, including two synthetic datasets `SynTab` and `SynImg`, and two real-world datasets VinDr-CXR (Nguyen et al., 2020) and e-SNIL (Camburu et al., 2018). `SynTab` contains tabular data, while `SynImg` and VinDr-CXR are image datasets, and e-SNIL consists of textual input. Figure 4 lists example of each dataset. Then, we focus on the design choices (Section 4.2) of our algorithm, and the performance under limited supervision (Section 4.3). Detailed experiment implementation can be found in Appendix C.

### 4.1 COMPARISON

We compare the overall performance of our proposed method with several baselines for both Level 1 and Level 2 tasks. The baselines used are as follows:

- Random: During testing, the exploration rate is set to $1.0$, i.e., the next action is selected randomly.

- DNN: This baseline trains a deep neural network with the same architecture as the DQN, but in a supervised manner. Instead of taking the state as input, it processes information from a single data sample $x_i$, where the input consists of the logits of the classifier for the predicted class $\hat{y}_i$, and its feature embeddings from extracted by applying its explanation as a mask over the input data $\psi(x_i, e_i)$, paralleling the state design $s$.

- AL (Active Learning): Active learning is used to select unlabeled samples for human review and labeling to improve model performance. In our comparison, we compare our method in selection with the query strategy Expected Gradient Length (EGL) in AL as given in (Settles, 2009). We selected EGL because EGL is based on model gradients, making it relevant for auditing settings: It helps search for errors that lead to incorrect predictions (Level 1 error) and use wrong features (Level 2) by using information about the network's internal representations. Therefore, EGL serves as a well-motivated and strong baseline for evaluating our algorithm's effectiveness.

To assess the performance, we measure the number of identified errors within a given $K$-step. $K$ is set to 50, 500, 1000 and 50 on the `SynTab`, `SynImg`, VinDr-CXR, and e-SNIL, respectively. Note that these $K$s are smaller than the number of all errors existing in the test set, i.e., $K$ is the upper bound for the errors detected in this experiment. We ensure that each sample is counted only once. For Level 2 errors, which we report them separately, we count only errors that exclusively belong to Level 2, ensuring no overlap with Level 1.

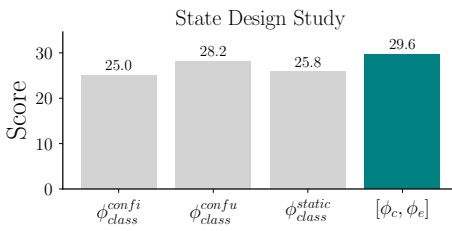

Figure 6: Comparison of different state designs. Light gray bars represent states without $\phi_{\text{explanation}}$.

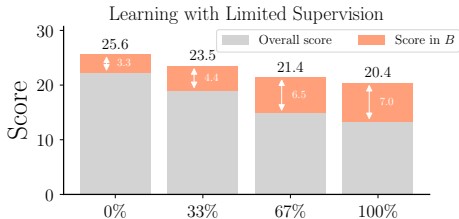

Figure 7: Comparison of scores when training with $p\%$ ($p \in [0, 33, 67, 100]$) of the labels for error type $B$.

Figure 5 presents a comparison of our method with baselines on fours datasets. Our RLAuditor outperforms the baselines on the first three datasets and achieves competitive results on e-SNLI compared to AL. Notably, it surpasses AL in detecting Level 2 errors significantly, which are more complex. Our method consistently outperforms the supervised baseline, demonstrating the superiority of our RL-based approach. This also supports our argument that formulating the problem as a sequential decision-making task is more appropriate, and that RL is a suitable solution.

## 4.2 STATE DESIGN

In this section, we study the different state designs $s$ in our RLAuditor algorithm. The intuition of the state is the information obtained from the viewed samples. Based on this information, an action $a$ will be taken. Therefore, the representation of the information from past samples is essential.

We first study the effectiveness of the component $\phi_{\text{class}}$ and its design proposed in Section 3.2, which we refer to "prediction **confidence** scores" denoted as $\phi_{\text{class}}^{\text{confi}}(i, j)$, with the other two designs:

- Prediction **confusion** scores: $\phi_{\text{class}}$ has the same dimension as the previous design, but we use the format of confusion matrix of all previously seen samples as the knowledge representation for the classifier. Each row corresponds to the ground truth class, each column represents the predicted class, and the entry at position $(i, j)$ represents the number of samples with ground truth class $i$ that were predicted as class $j$. Formally, $\phi_{\text{class}}^{\text{confu}}(i, j)$ is defined as:

$$\phi_{\text{class}}^{\text{confu}}(i, j) = |\{x_k \in \mathcal{U} \mid y_k = i \text{ and } \hat{y}_k = j\}|,$$

where $\mathcal{U}$ is the set of all previously viewed samples, $y_k$ is the ground truth label of sample $x_k$, and $\hat{y}_k$ is the predicted label of sample $x_k$ by the classifier.

- **Static** states: The two options above update $s$ after viewing one sample, i.e., $\mathcal{U}$ is dynamic. In this setting, we use the *static* state representation, which is a fixed representation. Similar to the prediction confidence representation, $\phi_{\text{class}}^{\text{static}}(i, j)$ is defined as:

$$\phi_{\text{class}}^{\text{static}}(i, j) = \frac{1}{|\mathcal{D}_i|} \sum_{x_k \in \mathcal{D}_i} \hat{y}_j^i,$$

where $\mathcal{D}_i = \{x_k \in \mathcal{D} \mid \hat{y}_k = i\}$ is the set of all test samples predicted as class $i$. Since $\mathcal{D}$ is fixed, the state remains static throughout the evaluation process.

Figure 6 shows the test scores using different state representations on SynTab in gray bars. Specifically, $\phi_{\text{class}}^{\text{confu}}$ achieves the best result. However, $\phi_{\text{class}}^{\text{confu}}$ requires the ground-truth class labels for $x_k$, which are not available in real-world scenarios. $\phi_{\text{class}}^{\text{static}}$ and $\phi_{\text{class}}^{\text{confi}}$ achieve competitive results, but relies on access to the full test set, which is impractical during deployment as new data comes. Considering these practical constraints, we select $\phi_{\text{class}}^{\text{confi}}$ as the final design choice for our algorithm.

We then study the effectiveness of $\phi_{\text{explanation}}$ in the state representation. Our final model is denoted as $[\phi_c, \phi_e]$. As shown in Figure 6, with the explanation information ($\phi_e$), our agent learns to detect errors most effectively, highlighting the importance of using model explanations.

## 4.3 LEARNING WITH LIMITED SUPERVISION

In this section, we aim to evaluate the performance of RLAuditor with limited training data for a specific error type (e.g., errors from a particular class, $B$). In this context, learning with limited

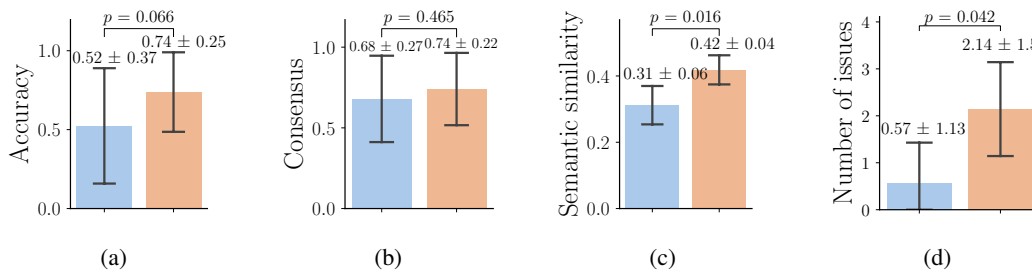

Figure 8: Human user study analysis among two groups: ▇ Control-; ▇ Experimental group. (a) Accuracy in identifying misclassifications. (b) Consensus in identifying misclassifications. (c) Participants' report similarity. (c) Number of issues identified in the model report.

supervision refers to identifying errors from $B$ with no or partial access to the annotated errors in $B$. We highlight this setting, because it is labor-intensive in practice to collect human annotations for model errors. We use `SynTab` for evaluation. During training, we annotate $p\%$ of the class $B$ ($B$ is randomly selected) and leave the remainder unannotated, with $p \in [0, 33, 67, 100]$. The results are listed in Figure 7. We report the total number of discovered errors across all error types (all classes) in gray bars, and the scores for the class $B$ in orange bars. We can see that with all the samples in $B$ are labeled ($p = 100$) during training, the number of the identified error for $B$ is the highest. However, without labeled errors in $B$ results in the highest total scores. This is because the algorithm finds it easier to learn when there are fewer error types. This experiment demonstrates that even with partially labeled data for an error type, our model can learn to detect it, highlighting its potential for efficient learning. More experimental results and detailed analysis can be found in Appendix D.

## 5 EVALUATION WITH HUMAN AUDITORS

One key challenge that RLAuditor aims to address is the scalability of practical audits. Effective auditing requires human-AI collaboration, as humans cannot scale alone and AI auditors need human judgment for accurate error labeling and learning domain knowledge. Thus, to inform future human-AI collaborative approaches to application auditing, we also conducted a human subject study to examine two research questions: **R1.** What strategies do human auditors use? **R2.** Can the samples selected by RLAuditor help humans better understand the model's behavior and errors?

Specifically, we implemented an auditing user interface (UI) that enables the study of interactions between human auditors and the RLAuditor. The dataset used for this task is a user intent dataset "Snips" (Coucke et al., 2018), which contains queries (short sentences) categorized into 7 intents. We finetuned a small pre-trained BERT variant model (Bhargava et al., 2021) to classify user intents, and this was the model to be audited. We adopted a between-subject design in which one group of participants was exposed to the RLAuditor-suggested samples (experimental group), while the other group was not (control group). $N = 12$ participants were recruited through institutional email lists and have experience in evaluating and debugging ML models. Both groups include PhD students in computer science, making them well-qualified for the ML audit task. The participants were randomly and evenly assigned to two groups, and compensated \$10 for their participation. Participants were tasked with summarizing model errors in an audit report, which included a set of question targeting Level 1 and Level 2 model errors. More details of the human user study can be found in Appendix E.

**F1: RLAuditor helps humans generate more accurate audit reports.** To assess the quality of the reports, we studied how participants identified different types of model errors. Participants in the experimental group were more likely to summarize misclassifications (Level 1 errors) and identify issues related to the model's feature usage (Level 2 errors). For each class, we asked participants to specify which other class(es) the model most frequently confused it with. Figure 8a presents the accuracy of misclassification identification (Level 1 errors), based on the ground-truth confusion matrix. The experimental group demonstrates higher accuracy. Figure 8b shows the pairwise Jaccard similarity within each group for the misclassification identification. The high consensus scores in both groups indicate strong agreement among participants on the identified errors. This suggests that the UI design influences the behavior of human auditors.

Regarding participants' analyses of model features, we observe that participants reported more similarly when using RLAuditor, as reflected in the report semantic similarity scores shown in Figure 8c. Participants summarized their observations on which features used by the model have issues or need improvement. With the help of RLAuditor, the experimental group demonstrated deeper reasoning and identified more issues (averagely two problems per class per user) related to the model's feature usage (Level 2 errors), as shown in Figure 8d.

**F2: RLAuditor helps humans complete audits in less time.** Participants were instructed to complete the task within 30 minutes but were allowed additional time if needed. We recorded the time each participant took to complete the task and analyzed the content of their reports. The average completion time for the control group was $37m55s \pm 6m45s$, while the experimental group completed the task in $32m26s \pm 4m38s$. The reduced time required by the experimental group highlights the effectiveness of the selected samples, as participants spent less time searching through the dataset. Notably, the efficiency gains are expected to become more as the dataset size increases. We also asked participants to rate how helpful they found the agent's suggestions in this audit task. The participants in the experimental group gave an average score of $5.75 \pm 0.5$ on a 7-point Likert scale, where 1 indicates "extremely useless" and 7 indicates "extremely helpful".

**F3: Human auditors rely on model explanations for audits.** In the participants' reported strategies, a common pattern emerged from both groups: $75\%$ percent of participants reported relying on model explanations (highlighted tokens) and summarizing insights by analyzing misclassification explanations. $17\%$ indicated they studied both correct and incorrect classifications in their audits. This aligns with prior work that addresses the usefulness of explanations in audit tasks (Yadav et al.; Balayn et al., 2022). It also validates the design of our RLAuditor, which incorporates model explanations to reflect human auditing strategies.

Based on findings **F1** and **F2**, we address research question **R2**: RLAuditor can assist humans in audits. The improvement comes from the agent clearly grouping problematic samples and presenting them. This allows humans to compare examples and better understand incorrect predictions or misused features. **F3** addresses **R1**, and it verifies that model explanations play an important role in application audits. More detailed analysis of the human user study can be found in Appendix E.

## 6 CONCLUSION

As ML models are deployed in real-world and safety-critical domains, auditing is essential to ensure their safe and responsible operation. In this work, we focus on application audits, which are important for assessing ML models but require domain-specific knowledge from diverse stakeholders and currently lack scalability. We formalize application audits as a sequential decision-making problem and introduce SAFAAI to guide the design of automated auditing algorithms. To our knowledge, this is the first work to model auditing as an MDP. We solve the MDP with an agent called RLAuditor that leverages RL and XAI methods to examine model behavior. We evaluate our proposed method across multiple datasets with various data modalities. Experiments demonstrate the effectiveness of our formalization and RL-based solution. This work provides a new way to address the auditing problem and support human auditors in human–AI collaboration for ML application audits.

**Limitations and Future Work.** Our work also motivates several directions for future work on auditing ML models, an important problem for AI safety. First, experimental results demonstrate that our RLAuditor can successfully automate parts of application auditing and complement human auditors but on traditional ML models. As foundation models are widely deployed in real-world applications, developing effective methods for auditing them becomes important. Since both SAFAAI and RLAuditor are model-agnostic by design, we believe that the insights developed in this work are directly relevant to the auditing of large foundation models. Nonetheless, achieving scalability will require further research into the design of expressive yet tractable state representations. Second, the RLAuditor focused on the first two levels of SAFAAI but not Level 3, which requires handling variability across tasks. To address this, in future work, we plan to extend the state representation with a meta-learning–inspired component to better capture task transitions. Lastly, our formalization should be viewed as one of several approaches to auditing, expanding the toolkit for ML auditing and providing a foundation to guide future research in this area. We emphasize the need for a deeper investigation into human-AI collaborative approaches to auditing. Auditing is a task that benefits from the complementary strengths of humans and AI in ensuring the responsible use of ML models.

**Ethical Statement**    In this work, we attempt to put human users at the center of human-AI collaboration, with the aim of designing an algorithm that can be efficient for facilitating humans in ML application audits. To safeguard user privacy and user rights, we have received approval from the university IRB. We believe that only when AI becomes more accessible, acceptable, and usable, can we realize its full potential to empower the world around us.

**Reproducibility Statement**    We have made every effort to ensure the reproducibility of our work. The paper provides detailed descriptions of the proposed algorithm in Section 3.2, with additional implementation details, hyperparameters, and experimental setups included in Appendix B. Full description of the datasets and experimental implementation details are presented in Appendix C. We also provide an anonymous link to the source code: `https://anonymous.4open.science/r/RLAuditor-F0F6/README.md`

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

APPENDIX CONTENTS

## A RELATED WORK

### A.1 METHODS FOR AUDITING ML MODELS

Recognizing the urgent need for the safe and responsible deployment of AI systems (Mökander et al., 2023; 2022), there is growing research focused on facilitating audits of ML models. Existing solutions can be broadly categorized as: (1) those designed to assist humans in performing audits and (2) those aimed at automating the auditing process. At the end of this section, we also discuss how model explanations can support and enhance the auditing process. Model audits are important not only for uncovering and fixing bugs in ML applications, but also for revealing hidden structures

of collaboration and advocating for a reshaping of existing practices in ML science (Lin & Jackson, 2023).

**Facilitating Manual Audits of ML Models.** In the first category, a common theme has been the development of application-specific user interfaces to explore model behaviors (Balayn et al., 2022; Wexler et al., 2019; Ren et al., 2016; Zhang et al., 2018). For instance, Wexler et al. (2019) introduced the What-If Tool, a model-agnostic interactive visualization tool designed to enhance model interpretability. Another strategy involves assisting humans in generating test cases by expanding the auditing dataset through generative models (Ribeiro & Lundberg, 2022; Rastogi et al., 2023; van Breugel et al., 2024). For example, Ribeiro & Lundberg (2022) propose a testing loop where a large language model (LLM) generates test cases, and human auditors review the top failing ones, adding valid tests to relevant topics or sub-topics to construct a comprehensive test dataset. While these tools provide valuable insights, the reliance on humans in the loop can limit their scalability.

**Automated Audits of ML Models.** Our work aligns with the second category of methods which focus on automatically detecting undesirable model behaviors (Kang et al., 2018; Ma et al., 2018; Singla et al., 2021; Lourenço et al., 2019). Examples of such methods include algorithms and frameworks for identifying problematic model behaviors in specific tasks, such as improving model accuracy through selective input sampling (Ma et al., 2018), rather than directly assisting human auditors. Kang et al. (2018) propose using a set of assertion rules to detect errors in machine learning outputs for object detection in videos. For example, a car should not disappear and then reappear in consecutive video frames. These assertions can be "soft" and represented as probabilistic. This probabilistic modeling approach is further extended to identify errors in human-provided labels and machine learning model outputs within the autonomous vehicle domain (Kang et al., 2022), where ensuring the accuracy of annotations is critical for safety and performance. Kang et al. (2022) propose a system that learns priors to distinguish likely from unlikely values. The resulting probabilistic model is then used to rank data point labels based on their likelihood of being erroneous. These existing automated approaches have primarily targeted domain-specific applications. In contrast, our method aims to provide a more general solution across diverse models and datasets.

**Role of Model Explanations.** Across both categories of methods, model explanations play a crucial role in audits by revealing how a model utilizes features, thereby helping auditors assess model behavior (Yadav et al.; Balayn et al., 2022). For instance, through a user study, Balayn et al. (2022) demonstrate the utility of diverse explanations in facilitating manual audits of ML models. In the context of automated methods, the XAudit approach (Yadav et al.) partially reconstructs a hidden model using model explanations, ensuring the inclusion of a specific feature. However, this method is constrained to simple models, such as linear regression models, and relies on predefined feature sensitivity metrics. In contrast, our work aims to leverage explanations to automatically audit more complex models and datasets, such as convolutional neural networks (CNNs) for image analysis. In this work, we select explanation techniques (SHAP and GradCAM) that are well-established and widely validated in the literature to ensure a reasonable level of supporting human-AI collaboration (Chandrasekaran et al., 2018; Chromik et al., 2021; Wang et al., 2022); see Table 3 in (Rong et al., 2023).

## A.2 LAYERED AUDIT FRAMEWORKS

Prior layered model audit frameworks such as Lam et al. (2024); Mökander et al. (2023) focus on high-level, organization-wide audits, often assuming the resources of large institutions and aligning with regulations such as NYC's Local Law 144 (Lam et al., 2024). In contrast, there has been comparatively less focus on application-level audits. However, these are increasingly important, as many ML model practitioners lack the capacity for large-scale, organization-level audits. They need practical, lightweight tools to conduct audits routinely within their specific applications.

Another line of audit frameworks focuses on defining metrics and evaluations that capture key aspects of fairness and interpretability. For example, Mitchell et al. (2021) define fairness using equal decision measures or causal reasoning, while (Doshi-Velez & Kim, 2017; Alvarez-Melis & Jaakkola, 2018) introduce metrics such as robustness of interpretability to assess model explanations. However, some measures, such as human-grounded evaluation of explanations, require manual effort and cannot

be fully captured by automated metrics (Doshi-Velez & Kim, 2017). These frameworks emphasize post hoc evaluations at the dataset or model level. For instance, Mitchell et al. (2021) evaluate social bias across the dataset, Alvarez-Melis & Jaakkola (2018) assess the robustness of all generated explanations and Doshi-Velez & Kim (2017) emphasize human understanding of a model by viewing some explanations. They fail to provide individual samples that human auditors may want to examine more closely.

Compared to these previous work, our SAFAAI serves a similar purpose by providing clear definitions that can be formulated as mathematical problems. However, our focus is on identifying individual error samples and presenting them to users to support the auditing process. This procedure is not mathematically modeled in prior works. The definition of "error" in our framework is general and can be determined by users or based on existing fairness or interpretability criteria.

### A.3 REINFORCEMENT LEARNING (RL)

RL is a general paradigm for solving sequential decision-making problems, including those involving uncertainty, through trial and error. In its most general setup, an agent learns to make optimal decisions by interacting with its environment, receiving rewards for its actions (Sutton, 2018). Through the history of trial and error, it refines a policy to maximize cumulative rewards. RL has been widely used in complex domains such as robotics, game playing, autonomous driving, and finance, where explicit programming of optimal behavior is impractical (Silver et al., 2016; Arulkumaran et al., 2017). In this work, we use RL to select the optimal next action (test case) for audits, while the development of new RL algorithms is not the main technical focus.

**Test Case Generation for Software using RL.** Prior work (Takerngsaksiri et al., 2025; Durmaz & Tümer, 2022; Kim et al., 2023; Kathiresan, 2024) has used RL to generate software test cases, which are modeled as sequences of actions such as input tokens, API calls, or UI interactions. These input sequences are designed to uncover bugs or test for malicious behavior in the system. For example, Takerngsaksiri et al. (2025) use deep RL within large language models to generate executable, high-coverage test code, while Durmaz & Tümer (2022) employ RL to generate minimal crashing input sequences. In these works, states represent the current program or input context, actions correspond to next steps, and rewards are based on outcomes such as code coverage or syntax validity.

In contrast, our work focuses on auditing ML models. The sequence of actions generated by our algorithm is not to trigger a system failure, but to construct a comprehensive understanding of the model's erroneous behavior: Unlike software debugging, where the goal is often to generate a failing test case, our method aims to select a representative set of test samples that reveal different facets of the model's failures to human auditors. Our state formulation simulates the accumulation of knowledge about model behavior, reflecting how humans build understanding through observations. This is a more challenging and cognitively aligned task.

## B IMPLEMENTATION DETAILS FOR RLAUDITOR

### B.1 RLAUDITOR: FORMALIZING EFFICIENT AUDITING AS AN MDP

We introduce **RLAuditor**: an automated approach to solving the *Efficient Application Auditing* problem. Our design is grounded in SAFAAI and informed by observations of how humans conduct application audits of ML models. Typically, auditors begin by evaluating model predictions and explanations on a few exploratory instances to develop an understanding of the model's performance and reasoning. They then exploit this understanding to more strategically select subsequent instances where model behavior is likely to be informative. As their familiarity with the model grows, they become increasingly effective at identifying its errors. Throughout this process, auditors must continuously balance exploration (i.e., learning more about the model's reasoning) and exploitation (i.e., using that knowledge to efficiently detect model errors). While this manual process may not scale well, it provides key insights for designing automated auditors. First, the sequential nature of selecting instances for model evaluation motivates formulating the auditing task as a *sequential decision-making problem*. Specifically, we model this process as a Markov Decision Process (MDP), which naturally captures the dynamics between the agent (the auditor) and the environment (the model's behavior), as well as the dependence on past observations. Second, the inherent *exploration-*

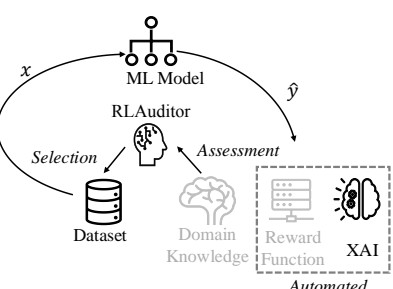

Figure 9: Auditing with RLAuditor.

*exploitation trade-off* in auditing motivates the use of *reinforcement learning* to address this challenge. Given an MDP formulation of the auditing process, we then learn a *policy for sequentially selecting the elements of set $C'$* (see Definition 3.2) using RL. Figure 9 depicts the process of learning this policy, using RL and human feedback. With model explanations, the utilization of features can be automatically assessed, further enhancing the auditing process. When deploying RLAuditor, the gray elements are excluded, reflecting the automated auditing process.

In practice, the expert is familiar with both the application and the model being audited, enabling them to identify errors. When working with RLAuditor, the expert's role goes beyond simple labeling. They provide feedback on whether selected samples are erroneous and can offer more advanced guidance, such as highlighting errors that RLAuditor may have overlooked. This feedback can be incorporated as a reward signal to refine and improve RLAuditor's performance. Moreover, in practice, the expert is responsible for composing comprehensive reports that summarize the different types of errors identified in the model.

## B.2 Implementation Details for Figure 2

Figure 2 illustrates the intuition behind modeling the auditing process as a sequential task, where knowledge about the model accumulates over time and the selection of subsequent samples is guided toward more informative errors. To validate this intuition in the implemented algorithm RLAuditor, we plot the training curve of RLAuditor on `SynTab`. The state learning error is measured using the mean squared error:

$$\mathcal{L}_{\text{MSE}} = \frac{1}{N * N} \sum_{i,j} \left( \phi_{\text{class}}^{\text{confi}}(i, j) - \phi_{\text{class}}^{\text{confu}}(i, j) \right)^2,$$

where $\phi_{\text{class}}^{\text{confi}}$ represents the learned state, and $\phi_{\text{class}}^{\text{confu}}$ denotes the ground-truth state, which requires the ground-truth labels (defined in Section 4.2). Note that it is impracticable to obtain the ground-truth state for $\phi_{\text{explanation}}$, so we omit its error estimation in $s$ for this plot. We also plot the reward curve over the training steps. The figure shows a clear trend: the estimation of $s$ improves over time, and the reward increases as the model knowledge aids error discovery. Both curves saturate after about 300 steps. This figure also shows that the selection of error samples is a sequential decision-making process that relies on $s$.

## B.3 Training Algorithm for RLAuditor

Algorithm for RLAuditor can be found in Algorithm 1. Our RLAuditor is adopted from Deep Q-Learning (DQN) (Mnih et al., 2015). DQN combines Q-learning with deep neural networks (DNNs) to enable agents to learn value-based policies directly from high-dimensional input. The algorithm trains an agent through techniques such as experience replay and a target network. To address a stable training outcome, Double DQN is used in Algorithm 1, which decouples action selection and evaluation by using the target network for value estimation.

---

**Algorithm 1** RLAuditor Training Algorithm

---

1: Initialize replay memory $L$
2: Initialize $Q$ with weights $\theta$
3: Initialize target $\hat{Q}$ with weights $\hat{\theta} = \theta$
4: Set the number of episodes $E$
5: Set the maximum number of steps per episode $T$
6: **for** $e = 1, 2, \ldots, E$ **do**
7:    $k \leftarrow 1$
8:    Select a random initial state $s_1$
9:    **while** Goal state not reached **and** $k \leq T$ **do**
10:       With probability $\epsilon$ select a random action $a_k$, otherwise select $a_k = \arg\max_a Q(s_t, a; \theta)$.
11:       Execute action $a_k$ and observe reward $r_k$.
12:       Set $s_{k+1} = T(s_k, a_k)$, and store transition $(s_k, a_k, r_k, s_{k+1})$ in $L$.
13:       Set $y_j = r_j$ if episode terminates at step $j+1$, otherwise $y_j = r_j + \gamma \max_{i \in \Omega(a_{k+1})} \hat{Q}(s_{k+1}, a_i; \hat{\theta})$.
14:       Perform a gradient descent step on $(y_j - Q(s_j, a_j; \theta))^2$ with respect to $\theta$.
15:       Update $\hat{Q} = Q$
16:    **end while**
17: **end for**

---

### B.4 IMPLEMENTATION DETAILS

$Q$ **Architecture.** The Q-network integrates state and action representations to estimate Q-values. It consists of two sub-networks. One maps input states to a 128-dimensional latent space using fully connected layers with ReLU activations. Actions are processed through another network with a linear layer, and ReLU, producing 64-dimensional action features. The model expands state features to match the number of actions, concatenates them with action features, and passes them through an estimation network comprising multiple fully connected layers and ReLU activations. The final output layer predicts a single Q-value per state-action pair. This architecture enables effective learning in environments with structured action spaces, making it suitable for complex decision-making tasks.

**Training Hyperparameters.** The training setup includes a batch size of 128 for processing training samples, and a replay buffer capable of holding up to 1M past experiences. The training begins after 50,000 steps, with updates occurring every four steps. The target network is updated every 1,000 steps. The learning rate is set at 0.001, with a smoothing factor of 0.95 applied for target estimation. Moreover, an exploration factor of 0.01 is used to balance exploration and exploitation during action selection.

## C IMPLEMENTATION DETAILS FOR EXPERIMENTS

### C.1 DATASETS

#### C.1.1 SYNTAB

We generate a tabular dataset consisting of five classes, with each class containing 1000 samples. The dataset includes ten features, and each class is sampled from a Gaussian distribution. To introduce Level 1 errors, we make two classes share a similar distribution by assigning them close values for the mean $\mu$ and standard deviation $\sigma$ in the Gaussian distribution. This can be observed in Figure 10, where class 0 and 1 are overlapping. For Level 2 errors, two out of the ten features are redundant, meaning they contain no class information. Using these two features for classification results in a Level 2 error. We randomly split 60% for $\mathcal{D}_{train}$, and the rest for $\mathcal{D}_{test}$. $\mathcal{M}$ is a trained linear regression model, and the feature embedding weights before classification, are used as the explanation $e$. Please note that $\mathcal{M}$ is trained on a larger, separate dataset rather than $\mathcal{D}_{train}$, where $\mathcal{D}_{train}$ refers specifically to the training set used for the RL-based algorithm for simplicity. This applies to all other datasets used in the experiments.

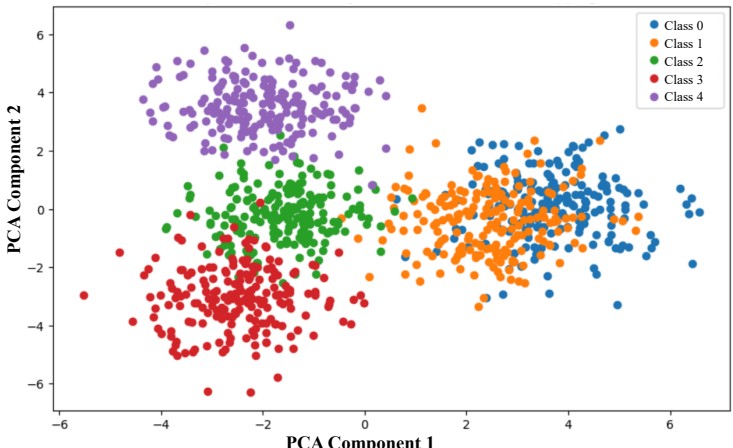

Figure 10: PCA analysis of classes in `SynTab`.

### C.1.2 SYNIMG

To further evaluate the performance of the proposed method on a more complex and larger dataset, we generate a challenging image dataset consisting of 24k images, with 12k for $\mathcal{D}_{\text{train}}$ and $\mathcal{D}_{\text{test}}$ each. This dataset is inspired by Yeh et al. (2020). Each image contains up to 15 shapes, with only five of them being relevant to the ground truth class.

Figure 11 lists the shapes that used in `SynImg`. We use the first five shapes ($q_i$) in Figure 11 to determine the five concepts ($c_i$) in the dataset. Concretely, we define concepts using specific formulas. The first concept calculates a binary outcome based on the formula:

$$c_1 = ((1 - q_1 \cdot q_3) + q_4) > 0.$$

The second concept computes the sum of $q_2$ and the product of $q_3$ and $q_4$:

$$c_2 = q_2 + (q_3 \cdot q_4).$$

The third concept sums two products:

$$c_3 = (q_4 \cdot q_5) + (q_2 \cdot q_3).$$

The fourth concept uses the `bitwise XOR` operation between $q_1$ and $q_2$:

$$c_4 = q_1 \oplus q_2.$$

Finally, the fifth concept adds $q_2$ and $q_5$:

$$c_5 = q_2 + q_5.$$

After obtaining the concepts, we encode different combinations of concepts into ten classes, i.e., each sample is defined by a combination of the five concepts. Specifically, we use the following algorithm: (1) It takes a list of 5-digit binary numbers representing concept combinations, and turns each binary number into a regular number (decimal). (2) It assigns a label by dividing that number by ten and taking the remainder. That way, all labels are between 0 and 9. Figure 12 shows examples from the dataset, where the it demonstrates the corresponding concept and class.

On this dataset, we train a five-layer CNN as the model $\mathcal{M}$ and use GradCAM (Selvaraju et al., 2017) for the explanation $e$. Figure 13 show more examples. It is straightforward to obtain annotations for Level 1 error by comparing the predictions with the ground truth labels. As for the Level 2, since the positions of the five relevant shapes are known during dataset generation, we can assess the correctness of the explanation $e$, i.e., whether the shapes corresponding to the ground-truth class are correctly highlighted by $e$ in the important areas based on a threshold.

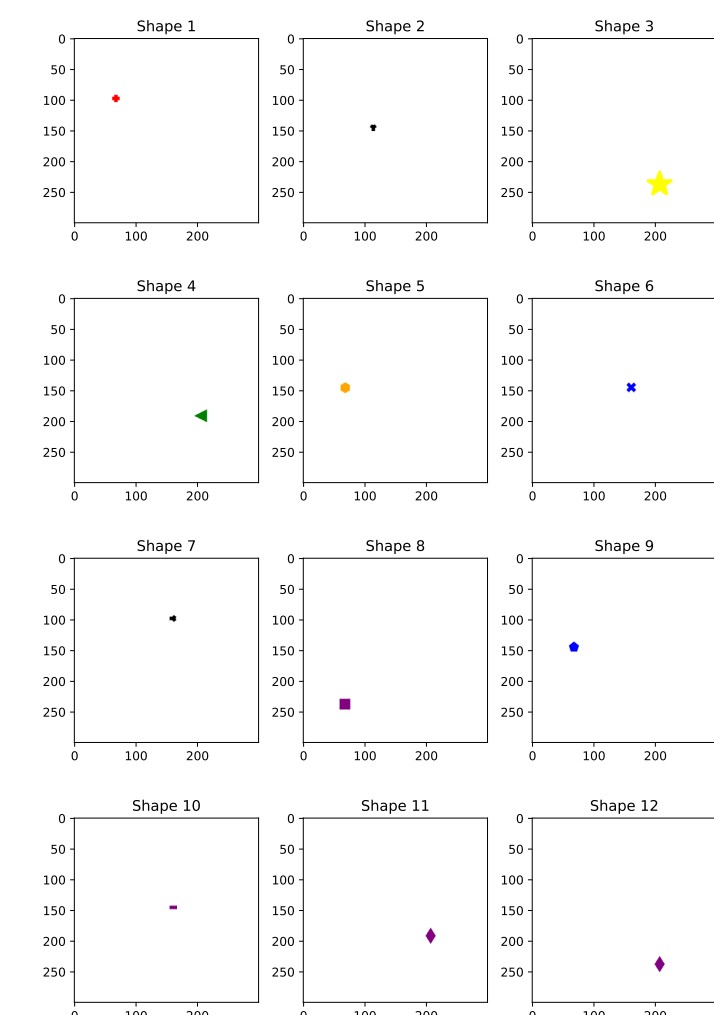

Figure 11: Shapes used in `SynImg`. The first five shapes contain class-discriminative information.

### C.1.3  VINDR-CXR

To verify the performance of our methods in real-world applications, we use a medical image dataset named VinDr-CXR (Nguyen et al., 2020). It consists of more than 100k raw X-ray images in DICOM format, annotated for the presence of 14 types of thoracic abnormalities, with each finding localized using a bounding box. We extract 8513 samples for $\mathcal{D}_{\text{train}}$ and 8654 for $\mathcal{D}_{\text{test}}$, ensuring each sample is assigned an abnormality label along with its corresponding bounding box. A ResNet-50 is used as $\mathcal{M}$. Similarly to `SynImg`, we identify Level 2 errors using Grad-CAM (Selvaraju et al., 2017) as the explanation $e$. A Level 2 error is determined based on whether the top-K attribution scores in $e$ achieve an Intersection over Union (IoU) with the bounding box exceeding a preset threshold.

In the real-world deployment of ML models in medical applications, Level 1 and Level 2 audits are critical for both patients and medical professionals. Level 1 is essential for patient safety, as incorrect predictions can lead to misdiagnoses. Level 2, on the other hand, is important for doctors, who must assess the accuracy of the model's decisions and determine whether to trust them. We use this dataset to demonstrate the practical application of our proposed SAFAAI and RLAuditor.

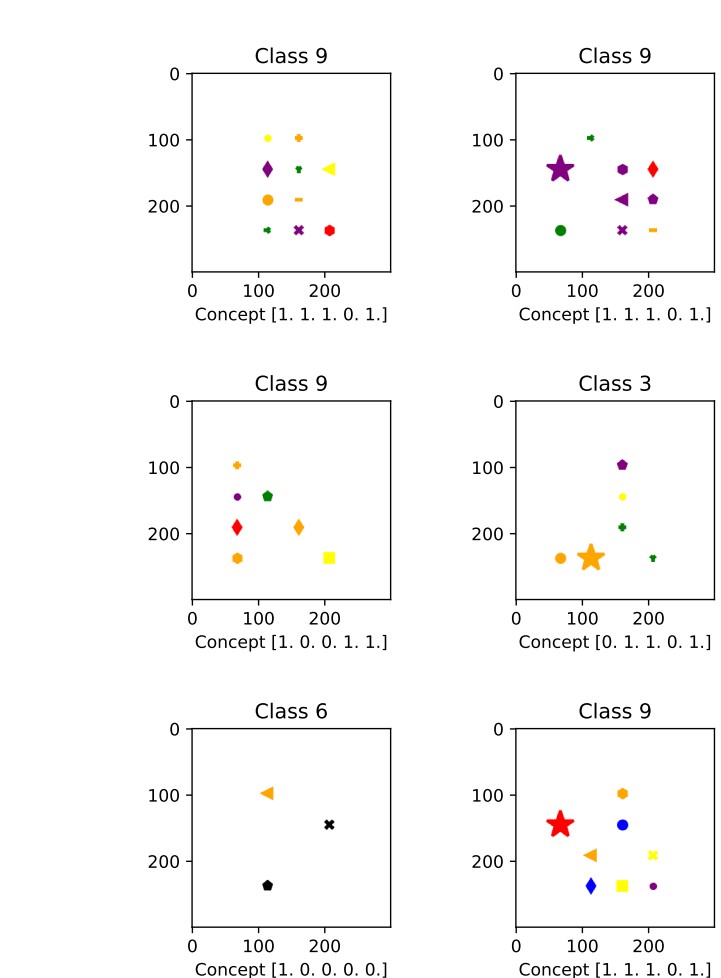

Figure 12: Examples from `SynImg`. Class and the concept labels are listed.

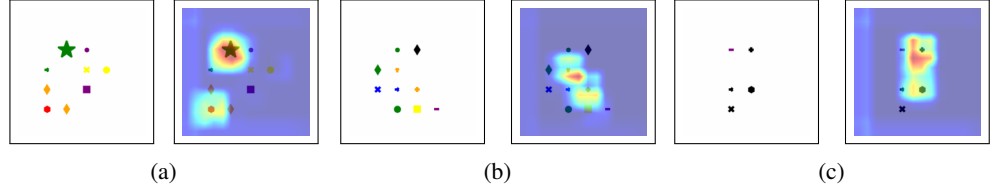

Figure 13: `SynImg`: Image $x$ and its explanation $e$.

### C.1.4 E-SNIL

The e-SNLI (Explainable Stanford Natural Language Inference) dataset (Camburu et al., 2018) builds on the original SNLI dataset (Bowman et al., 2015) to help make AI models easier to understand. A sample in the SNIL dataset contains a pair of sentences that is labeled as entailment (one follows from the other), contradiction (they disagree), or neutral (no clear connection). e-SNIL adds human-written explanations for why the pair has such a relationship and also highlights the words that are important

| Dataset | Val | Test |
|---|---|---|
| `SynTab` | 0.87 | 0.86 |
| `SynImg` | 0.83 | 0.82 |
| VinDr-CXR | 0.72 | 0.70 |
| e-SNIL | 0.89 | 0.88 |

Table 1: Accuracies of the trained model $\mathcal{M}$ used in experiments for audits.

for the decision (also annotated by humans). This makes AI systems more transparent and reliable. There are 9842 samples in $\mathcal{D}_{train}$ and $\mathcal{D}_{test}$ each.

On this dataset, we finetune a pretrained BERT model (Devlin et al., 2018) as $\mathcal{M}$, and use SHAP (Lundberg & Lee, 2017) as $e$. To obtain Level 1 error labels, we annotate the samples that are misclassified. For Level 2 error annotation, we first normalize the SHAP values within each sample and retain only the tokens with importance scores above 0.5. Next, we create a separate set of tokens from the human-highlighted words provided in the dataset. If the IoU between these two sets is below a predefined threshold, the sample is labeled as a Level 2 error.

## C.2 AUDITED ML MODEL

Table 1 lists the accuracies of the model $\mathcal{M}$ used in audits. Please note that the model is trained with a separate training dataset, which is different from the $\mathcal{D}_{train}$ used for training the RLAuditor. The size of the training set is 2k, 36k, 40.7k and 55k for `SynTab`, `SynImg`, VinDr-CXR, and e-SNIL, respectively (which is the original training set from the dataset). The validation set is in fact the $\mathcal{D}_{train}$ for training the RLAuditor, while the test set is $\mathcal{D}_{test}$.

## C.3 BASELINES

**Active Learning.** Our paper includes baselines based on active learning methods. Specifically, we adopt the Expected Gradient Length (EGL) (Settles et al., 2007) and compute it as follows:

$$x_{\text{EGL}} = \arg\max_x \sum_i^K f_\theta(\hat{y}_i \mid \mathbf{x}) \, \|\nabla \, l_\theta(\mathcal{L} \cup \langle \mathbf{x}, \hat{y}_i \rangle)\|, \tag{1}$$

where $f_\theta(\cdot)$ denotes the trained user model in our case with parameters $\theta$. To include $\mathbf{e}$ in the input, we use the explanation $\mathbf{e}$ as the weighted mask in the same manner as proposed to construct the feature embeddings $\psi(x_j, e_j)$ for the state $s$ in Section 3.2. $\mathcal{L}$ is the objective function for the model training, which is the cross-entropy loss. Let $\nabla \, l_\theta(\mathcal{L})$ be the gradient of the objective function with respect to $\theta$. The Euclidean norm of the objective function, $\|\nabla \, l_\theta(\mathcal{L})\|$ should be nearly zero since the model converged in the last round of training (Settles et al., 2007). Therefore, $x_{EGL}$ can be simplified as:

$$x_{\text{EGL}} = \arg\max_{\mathbf{x}} \sum_i^K f_\theta(\hat{y}_i \mid \mathbf{x}) \, \|\nabla \, l_\theta(\langle \mathbf{x}, \hat{y}_i \rangle)\|. \tag{2}$$

Running the EGL on all samples on the test set $\mathbf{D}_{test}$ and select the top-$K$ samples with the highest expected gradient values. Note that if the oracle labels $y_i$ are available, they are used in the EGL computation. However, in practical auditing scenarios, oracle labels are typically unavailable, so we instead rely on the predicted labels.

**DNN.** We utilize the deep neural network from the DQN algorithm (i.e., the target network) as described in Algorithm 1. Specifically, the DNN is trained on the same dataset $\mathcal{D}_{\text{train}}$, assigning a score to each state-action pair analogous to a reward signal. It is optimized using mean squared error loss with the reward $R(s, a)$ from the training set as the target. To ensure a fair comparison with the RL-based agent, we adopt the same training hyperparameters, setting the learning rate to 0.001 and the number of epochs to 200. After training, the DNN is evaluated on the $\mathcal{D}_{test}$.

| | SynTab | | SynImg | | VinDr-CXR | | e-SNLI | | Housing Prices | |
|---|---|---|---|---|---|---|---|---|---|---|
| | Level 1 & 2 | Level 2 | Level 1 & 2 | Level 2 | Level 1 & 2 | Level 2 | Level 1 & 2 | Level 2 | Level 1 & 2 | Level 2 |
| Random | $11.4 \pm 2.3$ | $3.5 \pm 1.8$ | $238.8 \pm 10.9$ | $188.8 \pm 11.0$ | $360.4 \pm 13.9$ | $61.8 \pm 5.6$ | $14.0 \pm 2.4$ | $8.6 \pm 2.2$ | $27.5 \pm 15.3$ | $1.6 \pm 0.5$ |
| DNN | $21.5 \pm 2.3$ | $5.0 \pm 0.3$ | $386.0 \pm 3.7$ | $0.0$ | $438.5 \pm 5.5$ | $0.0$ | $23.6 \pm 2.5$ | $13.2 \pm 1.4$ | $33.1 \pm 3.3$ | $11.0 \pm 1.5$ |
| AL | $28.0 \pm 0$ | $1.0 \pm 0$ | $320.0 \pm 0$ | $193.0 \pm 0$ | $490.0 \pm 0$ | $67.0 \pm 0$ | $28.0 \pm 0$ | $12.0 \pm 0$ | — | — |
| W/o learned states | $28.9 \pm 1.5$ | $6.4 \pm 1.5$ | $431.9 \pm 11.9$ | $328.8 \pm 18.4$ | $876.7 \pm 6.4$ | $152.3 \pm 8.8$ | $24.8 \pm 0.7$ | $16.0 \pm 0.6$ | $35.6 \pm 4.1$ | $15.3 \pm 1.7$ |
| W/o expl | $26.7 \pm 2.2$ | $5.4 \pm 1.1$ | $388.3 \pm 4.2$ | $210.2 \pm 9.0$ | $873.7 \pm 7.6$ | $160.2 \pm 19.5$ | $23.4 \pm 0.6$ | $14.6 \pm 0.5$ | $33.8 \pm 2.7$ | $15.2 \pm 2.1$ |
| **Ours** | $29.6 \pm 1.3$ | $9.0 \pm 0.8$ | $458.3 \pm 3.2$ | $336.8 \pm 2.7$ | $885.5 \pm 7.2$ | $163.1 \pm 6.8$ | $27.3 \pm 0.9$ | $17.0 \pm 0.9$ | $37.2 \pm 4.7$ | $17.0 \pm 1.6$ |

Table 2: Comparison of auditing results for two levels of errors across four datasets. Each dataset's $K$ is set to the same number for all methods. The number of identified errors is listed.

# D  MORE EXPERIMENTAL RESULTS

In this section, we present additional quantitative results.

## D.1  FULL RESULTS OF COMPARISON

First, we report the complete results, including the mean and standard deviation on five independent runs, as shown in Table 2. The results for the first four datasets in Table 2 complement those presented in Figure 5 of the main paper.

We also include the two additional baselines compared to Figure 5:

- W/o learned states: This baseline does not utilize the states learned from the training set (assuming that the training and test sets share a similar data distribution), but initializes the state to zeros at the start of testing.
- W/o expl: This baseline does not include $\phi_{\text{explanation}}$ in the state $s$, although it still contains the labels for Level 2 errors during training.

It is worth noting that when the explanation features are not in the states, the detection of Level 2 errors is worse compared to our method. Moreover, if learned states are not used during testing, performance declines but the RL algorithm can still learn knowledge from the observed samples.

## D.2  EXTENSION TO REGRESSION TASKS

Section 4 mainly discusses RLAuditor performance on classification tasks. However, our algorithm can be extended to regression tasks with two main adaptations in: (1) constructing $s$ and (2) designing the reward $R(s, a)$. In the original formulation, $s$ is defined based on class information, which is not available in regression tasks. To address this, we manually partition the samples into $N$ bins according to their predicted values. Once $N$ is fixed, each sample can be assigned to a bin $i$, viewing it as a class. To obtain $\phi_{\text{bin}}$, we get the feature embeddings before the regression prediction, and thus

$$\phi_{\text{bin}}(i, k) = \frac{1}{|\mathcal{U}_i|} \sum_{x_j \in \mathcal{U}_i} \psi_k(x_j),$$

where $\psi_k(x_j)$ denotes the feature embedding of dimension $k$ for the sample $x_j$. On the other hand, the information for $\phi_{\text{explanation}}$ is constructed from the explanations $e$ as

$$\phi_{\text{explanation}}(i) = \frac{1}{|\mathcal{U}_i|} \sum_{x_j \in \mathcal{U}_i} e_j,$$

where $e_j$ denotes the explanation for sample $x_j$. In this setting, we use $e_j$ directly as features rather than as a mask. This is because masking could distort the regression prediction, and explanations in regression tasks do not have spatial structure. Therefore, they can be averaged across each dimension.

To design the reward for regression tasks, we first compute the absolute prediction error for each sample as

$$\text{error}_j = |\hat{y}_j - y_j|,$$

where $\hat{y}_j$ is the model prediction and $y_j$ is the ground-truth label. To emphasize larger errors, we introduce an exponent parameter $\alpha > 1$ and define the reward as

$$R_j = (\text{error}_j)^\alpha.$$

In this formulation, $\alpha$ controls the degree to which large errors are penalized. $\alpha > 1$ amplifies the contribution of larger errors. We set $\alpha = 2.5$. This design encourages the algorithm to prioritize samples where the model performs poorly, focusing learning on selecting samples with high uncertainty and predictive error.

**Dataset and Audited ML Model.** We use California House Price dataset (Pace & Barry, 1997). It includes property attributes, geographic information, and sale prices, and is used to analyze factors affecting home values and to build models for predicting future house prices. The dataset contains 20,640 samples, which are split into 12,384 for training, 4,128 for validation, and 4,128 for testing. The training set is used to fit a Random Forest regression model, which is the model to be audited. The model is configured with 200 estimators, no maximum depth restriction, and parallel processing to improve efficiency. The trained model achieves a test RMSE of 0.509 and an $R^2$ score of 0.811, indicating strong predictive performance on unseen data. When constructing the states for RLAuditor, the data is divided into bins based on quantiles, meaning each bin contains roughly the same number of values rather than equal numerical ranges. This ensures that the distribution is represented evenly across all bins, even if the data is skewed. We set $N$ to 500. In practice, we found that a smaller $N$ results in overly coarse categorization, which can reduce performance. We compute SHAP values to provide explanations. The level 1 error is estimated using $R$ as described earlier. For the level 2 error, we label the top-10 most extreme feature importance scores for each feature as incorrect.

During training RLAuditor, the validation set is used, and the performance of the trained algorithm is evaluated on the test set. Table 2 lists the results of detected number errors within $K = 50$ steps. Note that AL requires gradient computations, which are not applicable to Random Forest models. Thus, this comparison is omitted from the table. Our algorithm continues to outperform the other methods, demonstrating that our algorithm also generalizes to regression tasks.

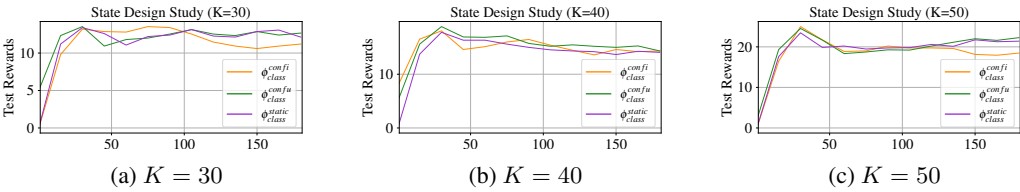

(a) $K = 30$      (b) $K = 40$      (c) $K = 50$

Figure 14: Comparison of test rewards when training with different state designs. The average test rewards over 15 episodes are shown. Three values of $K$ (number of selection attempts) are evaluated to the effect of the state design.

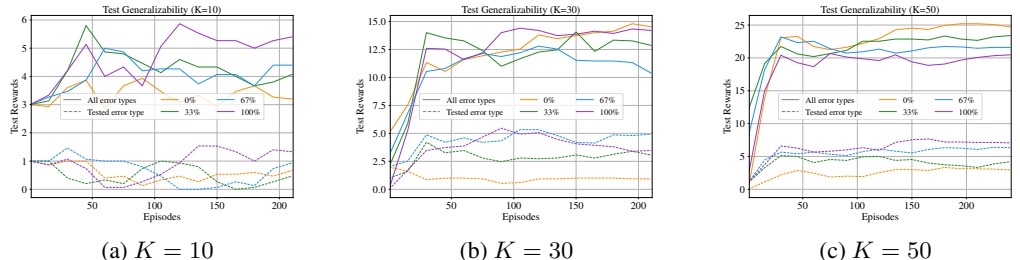

(a) $K = 10$      (b) $K = 30$      (c) $K = 50$

Figure 15: Comparison of test rewards when training with $p\%$ ($p \in [0, 33, 67, 100]$) of error type $B$. The average test rewards over 15 episodes are shown. Three values of $K$ are used.

### D.3 EXTENDED FIGURES

Figure 14 provides extended figures for Figure 6 in Section 4.2. We evaluate the performance under various values of $K$, and we can see that $\phi_{\text{class}}^{\text{confu}}$ consistently achieves the best results across all three settings. However, $\phi_{\text{class}}^{\text{static}}$ performs comparably when $K = 30$ or 50. Despite their strong performance, both settings have limitations in real-world applications. In contrast, while $\phi_{\text{class}}^{\text{confi}}$ is inferior due to

limited access to additional knowledge, its performance only marginally decreases, demonstrating the robustness and quality of this design.

In Figure 15, more results on running with $p\%$ errors type $B$ are shown. We use $K = 30$ and 50. From the result, we observe that with all the samples in $Y'$ are labeled ($p = 100$) during training, the number of the identified error for $Y'$ (represented by the purple dotted line) is the highest among the three different values of $K$. As fewer in $Y'$ are labeled, fewer errors are discovered. For example, in Figure 15c, the number of identified errors decreases as fewer samples are labeled. However, the best overall number of identified errors depends on the value of $K$. For instance, when $K = 50$, the setting with no labeled errors in $Y'$ results in the highest total test reward, outperforming the setting where all samples in $Y'$ are labeled, as shown in Figure 15c. This is because it is easier for the algorithm to learn from a single error type.

Our RLAuditor on `SynTab` fails in scenarios when there is a nuance in the prediction distribution between two classes with fewer samples for the agent to learn from (e.g., class 4 and class 3, as shown in Figure 10), this can present a challenge for the agent. One potential failure case that our current method cannot detect involves using incorrect features that are not among the top-k most important features. Since the algorithm focuses on the top-k features, it may ignore these "non-dominant" features, which could lead to problems in safety-critical applications where certain features are strictly "forbidden." However, the reward function can be adjusted to address the concern.

### D.4 Training Efficiency

In this section, we discuss the efficiency of the training of our RLAuditor. First, we elaborate the convergence of the training. Convergence means that the agent achieves a stable performance after training several epochs. From the learning curves in Figure 15 and Figure 14, we see that our model achieves a relatively stable score after 200 epochs. To evaluate its convergence, we report the performance of our trained agent by (1) averaging results over multiple consecutive episodes (15 episodes) and (2) plotting the reward curves in our experimental results. Furthermore, we utilize techniques such as Double Q-Learning, and an epsilon-greedy strategy during training, which generally enhance the convergence of the agent. The converged training curve suggests that our reward design is sufficient to effectively guide learning.

Once the agent is trained, the inference process is highly efficient, as it only requires a simple feedforward pass through a DNN, given that we are using DQN. Specifically, we measured the training time on the SynTab dataset, where our method took approximately times longer ($519s$) compared to the supervised learning (SL) method ($136s$). However, the inference times for both methods are comparable, as both require a feedforward pass through the NN to compute the sample score (or label). Moreover, training efficiency is reflected in limited supervision, as analyzed in Section 4.3. With limited labeled data for error type $B$, our method still achieves reasonable performance in detecting errors.

## E Evaluation with Human Auditors

We developed a user interface to enable the study of interactions between human experts and the proposed RLAuditor. The user study aims to address the following two research questions:

- R1: Can the samples selected by RLAuditor help users better understand the model's behavior and errors?
- R2: What strategies do human experts employ during audits?

In this section, we will introduce implementation details, including the model and the dataset used, followed by the user study details such as the procedure and the participant demographics, and additional analysis of the user study.

### E.1 Implementation Details

**Dataset and ML model.** The Snips dataset serves as a benchmark for intent classification and slot-filling in spoken language understanding (Coucke et al., 2018). It includes crowdsourced queries

covering seven distinct intents, including "PlayMusic," "BookRestaurant," and "GetWeather." The dataset is commonly used to assess natural language understanding models. It consists of 13.1k queries for training and 1.4k for testing. We finetuned a small pre-trained BERT variant model (Bhargava et al., 2021) using the training set as our model to be audited $\mathcal{M}$. The explanation $e$ is computed by the SHAP algorithm. We then use the subset from the test set as $\mathcal{D}_{train}$ to train our RLAuditor. To ensure the study was comprehensible for participants, we limited the audit dataset to 200 samples drawn from the test set. The trained model's accuracy on the test set is $89.5\%$.

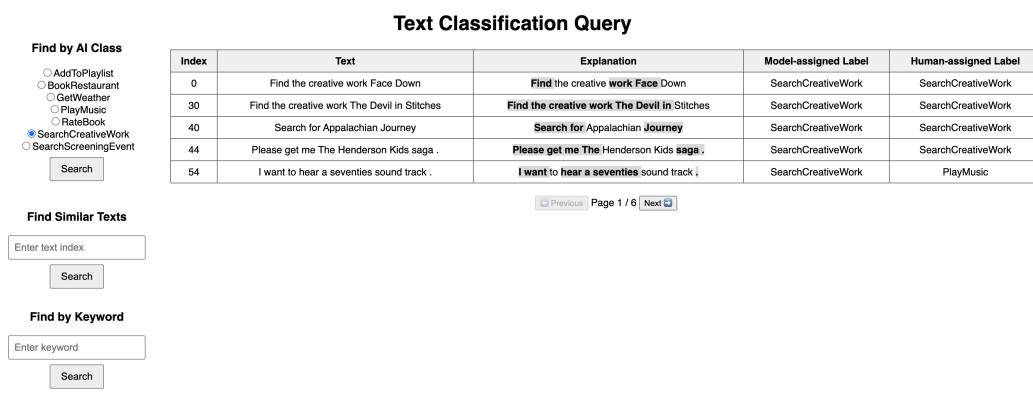

Figure 16: UI for the control group.

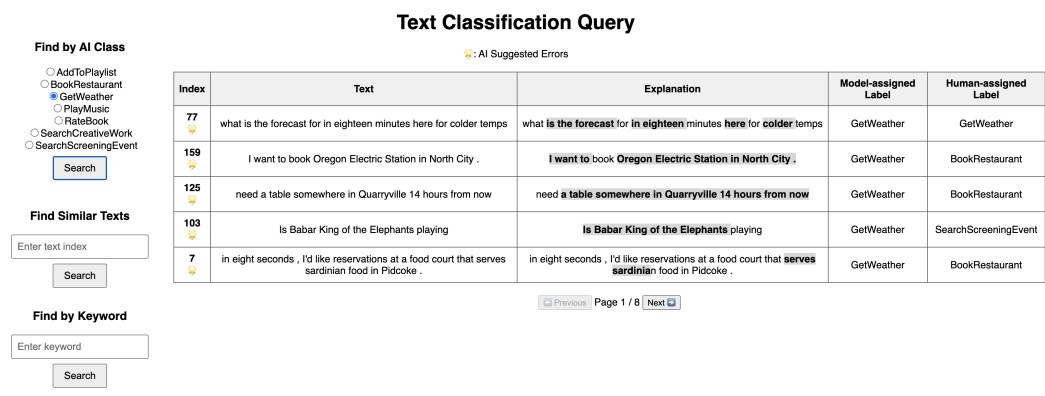

Figure 17: UI for the experimental group.

**UI.** Figure 16 and Figure 17 demonstrate the UI interface for each group. In this user interface, we provide three filtering techniques to assist users in exploring the data. First, samples can be categorized based on the class predicted by the model $\mathcal{M}$; this option is available in the top-left corner under the label "Find by AI Class". The second feature, "Find Similar Texts", allows users to investigate the model's behavior on inputs similar to a specific entry. By entering the index of a sentence, the system returns a list of texts ranked by cosine similarity between the queried sentence with the other test samples. he third function enables users to search for entries containing a specific keyword, offering a direct way to examine how the model handles certain terms or topics.

For each test sample, the user interface displays the original text, a visual explanation highlighting important words based on SHAP values, the model-assigned label, and the human-annotated (ground truth) label.

The key difference is that the experimental group was presented with samples recommended by our RLAuditor, which were highlighted and positioned at the top of the table.

### E.2 USER STUDY PROCEDURE

This user study is approved by the IRB (Institutional Review Board). We recruited 14 participants from the university, all with backgrounds in engineering and general machine learning. Two of these participants conducted a pilot study to assess the clarity of the experiment procedure and evaluate the time constraints involved. The final procedure for the user study is as follows:

- Participants read the task description and receive an explanatory introduction of the UI to get familiar with the UI.
- Participants start to formulate their **report** with the help of the questions proposed.
- Participants answer the **objective questions** of model behaviors.

After the analysis task of the application audits, participants were asked to fill out a questionnaire about their experience interacting with the UI.

The whole session takes 40 minutes and each participant received a $10 gift card as compensation. We conducted a between-group human experiment study. 6 participants in the control group did not receive the AI-suggested samples, while 6 participants in the experimental group did.

### E.3 PARTICIPANT DEMOGRAPHICS

All participants were recruited via an institutional mailing list. Each has experience in evaluating and debugging machine learning (ML) models in applied settings and holds at least a bachelor's degree in computer science. The cohort includes PhD students, PhD candidates, and one postdoctoral researcher, making them well-qualified to perform this challenging task.

In the control group, there are two females and four males, with a mean age of $27.8 \pm 4.2$ years and an average of $5.3 \pm 1.2$ years of experience in ML. On a 5-point scale, they rated their familiarity with evaluating ML models at an average of $4.2 \pm 0.7$. The experimental group comprises five males and one female, with a mean age of $27.2 \pm 2.2$ years and an average of $5.6 \pm 2.6$ years of ML experience. Their self-reported familiarity score averages $3.6 \pm 1.1$.

### E.4 GUIDING QUESTIONS FOR AUDITING

For each of the seven intent classes, we use the following questions to guide participants to formulate their audit report:

**Misclassification Pattern**

- Which other class(es) does the Model most frequently confuse with this class? (Multi-select problem)
- Why do you think these misclassifications occur? (Briefly explain why. With examples if possible.)

**Feature Attribution**

- Which features (e.g., specific words or phrases) does the Model rely on most when classifying this intent? (Give keywords.)
- Are these features appropriate and meaningful for this intent, or are they misleading/problematic? (Explain your judgment.)

They answer these questions for all seven classes in the dataset.

### E.5 OBJECTIVE QUESTIONS

At the end of the study, we also distributed a questionnaire to gather participants' feedback on their experience. The responses were used as follows:

1. How easy or difficult did you find the Model Auditing task? (1-Very Easy, 7-Very Difficult)

2. What aspects of the Model Auditing task did you find challenging, if any?

3. How did you conduct the Model Auditing task? (e.g., your strategy, functionality of the Auditing UI you found useful)

4. What additional information do you think you need for the Model Auditing task? (e.g., information about the dataset, potential features of the Auditing UI)

5. Do you think AI algorithms could assist you in this model auditing task? (Yes, No, or Unsure.)

6. Please share any additional comments or feedback about the task or study.

The second last question is designed for the experimental group.

### E.6 EXTENDED ANALYSIS

**Analysis of reports.** When studying the model features observed by participants, the experimental group tended to agree on more meaningful and relevant features, as illustrated in Figure 19. For instance, for the class *Search Screening Event*, the experimental group referenced both verbs and nouns—such as "find," "want," "see," and "would like", as the control group focused more on nouns indicating screen programs. In another example *Rate Book* shown in Figure 18, both groups selected the words "rate" or "give" as the most important tokens, while the experimental group also pointed out the words "stars" or " points", which are also highly relevant to rating.

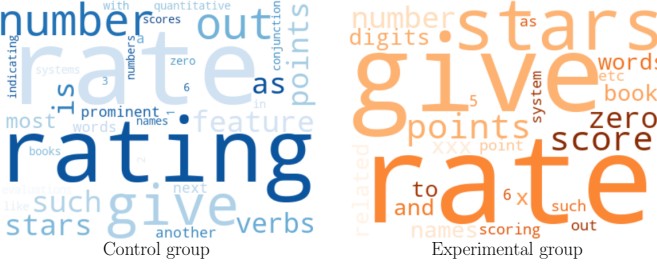

Figure 18: Word cloud in the model explanations used by the model for the class "*Rate Book*".

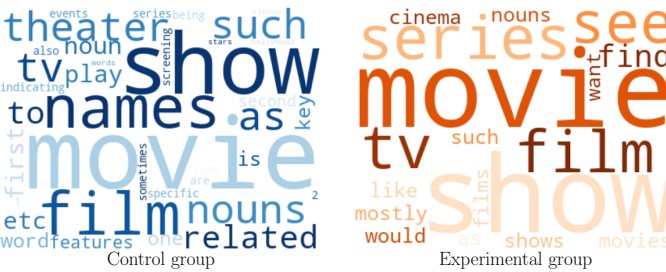

Figure 19: Word cloud in the model explanations used by the model for the class "*Search screening event*".

**Analysis of objective questions.** For the 1. objective question, the control group gives a score of $3.8 \pm 1.7$, while the experimental group gives a score of $3.0 \pm 1.4$. The difference between the groups was not statistically significant, with a $p$-value is $0.5$. However, it shows that the participants with the assistant of RLAuditor rate the auditing task as easier than the control group.

For the second question, participants expressed various challenges. For example, one from the control group noted, *"It was challenging to answer questions for specific categories without having reviewed the rest of the categories."* A holistic and comprehensive view is essential for auditing. However,

obtaining such an overview becomes increasingly impractical for humans as the dataset size grows, as another participant from the control group stated *"Checking all the classes for all possible mistakes."* We believe that human-AI collaboration is a promising way to solve this issue. Furthermore, two participants in the experimental group claimed that identifying features used by the model is difficult.

Regarding the 4. question, it is worth noting that one participant in the control group requested a "better filter/ranking mechanism." This suggests that human auditors also appreciate some form of pre-filtering, which can be effectively provided by an algorithm. This feedback further supports the usefulness of our proposed algorithm. One participant in the experimental group suggested including the model's confidence alongside the AI-suggested errors. Moreover, the importance scores for each token in the explanations were also noted as useful. The feature allowing users to view samples by predicted class was utilized by all participants, while the other functions were rarely used. This suggests that for future auditing UI design providing a clear and well-organized view including classification statistics is critical.

In the general feedback on our study (last question), two participants agreed that more samples should be displayed, preferably on a full page. This suggests that human auditors benefit from reviewing a larger number of examples at once to gain a comprehensive understanding of the model's behavior.

## F    COMPUTATIONAL INFRASTRUCTURE DETAILS

All experiments in this paper are conducted on the device given in Table 3.

Table 3: Computational infrastructure details.

| Device Attribute | Value |
|---|---|
| Computing infrastructure | GPU |
| GPU model | NVIDIA A40 |
| GPU number | 1 |
| CUDA version | 12.3 |

## G    STATEMENT ON LLM USAGE

Large Language Models (LLMs) were only used for minor grammar and language polishing. No part of the research ideation, experimental design, or manuscript content was generated or influenced by LLMs. The authors retain full responsibility for the scientific content and quality of the paper.

