# OpenReview forum: "Formalizing Audits of ML Models as a Sequential Decision-Making Problem"
_ICLR.cc/2026/Conference — ICLR 2026 Conference Withdrawn Submission_

### Official Review · Reviewer_2LNa · 2025-10-29

**Soundness:** 1
**Presentation:** 2
**Contribution:** 1
**Rating:** 2
**Confidence:** 4

**Summary:**

This paper proposes to approach ML classifiers audits with reinforcement learning, for discovering zones where they make the most errors.
It claims to be the first to formalize audits as a sequential decision-making problem.

**Strengths:**

* simple application of a RL scheme for finding next possibly error-leading inputs for classifiers

The topic of auditing machine learning and decision-making systems is important.

**Weaknesses:**

* unclear placement of the proposal concerning recent related works (e.g. active learning applied to auditing [2]), or current trends of audits to watch for bias (besides errors only)
* the setup if very restrictive, far from a more widely usable black box approach (requires predictions for all classes, features attribution scores, etc)
* applies rather standard RL without a contribution that would have come from the problem setting; in that light, this is more of an empirical paper only
* claims the introduction of a 3-part framework (SAFAAI), but only addresses the first two parts

More details:

I find the paper unclear in its presentation of what it claims (w.r.t. what is really addressed) as compared to the current state of the art of audit techniques this paper wants to overcome with a first formalization.

* a motivation for the current paper is that audits are currently "manual". But there are many audit techniques that rely simply on choosing a metric (eg, fairness with disparate impact) and consist in sampling the classifier input space to see if it goes over a predefined threshold for some sub-populations (see eg [1]), I fail to see how this is very manual, yet it serves the purpose described in the introduction of the paper in my opinion.

* it is also unclear to me how the authors place their work with regard to active learning applied to auditing (see eg [2] and the many papers citing it). Here also, the decision for the next sample is taken based on the previous ones. There is a full body of work on that matter, that is ignored from the paper discourse and related work (in particular Section 2, "automated audits"). The listing of active learning in the experiments as a baseline arrives too late and is not conclusive in the differentiation.

* the introduction/motivation is very generic. There are now so many papers in the related work, that new papers must in my opinion be more focused. For instance, Figure 2 is a totally generic plot of the evolution of a RL-based scheme: there is no indication or explanation to which data/problem it corresponds to in the paper, and could actually have been taken as in from any introductory book on RL.

* the operational setup for the audits only becomes clearer while reading the technical part; we are far from a widely applicable black or grey-box approach, as the requirements for the MDP proposal to function are high in practice. For instance, while here it is assumed that the auditor has access to feature attribution scores, some related works are proposing ways to infer which features are at use [3]. I get that we are here more in a white box setup, but the disconnection with what the community is assuming these days as a setup to define where audits can go is too important. And must be motivated precisely.

* Fig 4 is totally unreadable, and not explained in the main paper, so I do not get the value of it at this stage

* Section 5, that introduces the notion of collaboration with humans for audits, arrives as a surprise, and was not properly claimed in the contribution. This again blurs the whole message of the paper.

In general, I think the topic is important, but I have the feeling that authors had as a starter an RL approach to attack this MDP scheme, and then constructed a motivation for the whole paper (cf Fig. 2), but it is to far away from state of the art to be impactful in my opinion, so I would encourage authors to clarify their position and claims.

[1] Cherian, J. J., & Candès, E. J. (2024). Statistical inference for fairness auditing. Journal of machine learning research, 25(149), 1-49.

[2] Yan, T., & Zhang, C. (2022, June). Active fairness auditing. In International Conference on Machine Learning (pp. 24929-24962). PMLR.

[3] Rastegarpanah, B., Gummadi, K., & Crovella, M. (2021). Auditing black-box prediction models for data minimization compliance. Advances in Neural Information Processing Systems, 34, 20621-20632.

**Questions:**

None

---

### Official Review · Reviewer_etWa · 2025-10-31

**Soundness:** 3
**Presentation:** 2
**Contribution:** 3
**Rating:** 6
**Confidence:** 2

**Summary:**

Focusing on application audits, this paper proposed a framework, SAFAAI, which defines the objectives for this problem. And it proposed a novel RL-based auditing method, RLAuditor, to automate application audits.

**Strengths:**

1. The RL-based auditing method effectively picks up test samples that the target model will misclassify. Its performance surpasses baseline methods on four datasets.
2. It explores different state designs and provides evidence for the advantage of the selected scheme.
3. The user study confirms that RLAuditor can help human auditors.

**Weaknesses:**

1. Definition 3.2 of Efficient Application Auditing Problem focuses only on maximizing the number of detected errors but ignores the FPR of detection.
2. While for images, GradCAM automatically applies to test samples, the explanation for test E-SNL data relies on human annotation. This reliance is not practical, as such human intervention violates the motivation of RLAuditor and compromises its contribution.
3. In Lines 230-242, the definition of e and $\psi$ needs to be clarified. I don't understand where they're derived from and how you compute the function.
4. RLAuditor does not achieve the third-level objective in the SAFAAI framework.

**Questions:**

Can the proposed method scale up to other large-scale image datasets?

---

### Official Review · Reviewer_cUsg · 2025-10-31

**Soundness:** 1
**Presentation:** 3
**Contribution:** 1
**Rating:** 2
**Confidence:** 4

**Summary:**

This paper explores the use of RL to improve "auditing", understood here as "application auditing", and implemented as finding as finding prediction errors in classifiers.

**Strengths:**

-clear writing, easy to follow
-important topic: auditing ML models is important, especially in contexts where few labels are available to the auditor

**Weaknesses:**

-unclear model: I do not understand the precise usage scenario envisioned by the authors.
While l.54 it seems that the end-users perspective is taken -- which would be coherent with a rather black box perspective, we read that these end-users are supposed to routinely conduct those audits, submit reports, and have access to a sizable D_train dataset.
Given then that the objective (identify misclassification) is pretty far from many traditional black box audit approaches (like assess model fairness), I then tend to believe the considered auditors are rather employees of the model provider. Which leads to the question: why don't they use more "white box" approaches that would leverage the internals of the model to better identify model weaknesses ?

-narrow view of auditing that leads to unsupported claims. Auditing is far more than finding model misclassifications. The title "formalizing audits of models as a sequential decision making.." implicitly ignores large branches of auditing that already explored active approaches, RL-based approaches. See e.g. "Basu, Debabrota, and Udvas Das. "The Fair Game: Auditing & debiasing AI algorithms over time." Cambridge Forum on AI: Law and Governance." or "Online Fairness Auditing through Iterative Refinement" that imho definitely call for nuance in the first paper contribution.

Moreover, if one considers the practical implementation of auditing provided in the experiments (ie find misclassifications), a whole body of research that stems from software testing is ignored. See e.g. "Abo-eleneen, A., Palliyali, A., & Catal, C. (2023). The role of Reinforcement Learning in software testing. Information and Software Technology, 164, 107325.". Arguably, assuming model developers reply to the envisioned audit reports by proposing a new model version, the related works scope end up spanning the whole active learning field.

-Interestingly, in software testing a central notion is "coverage": the number of misclassifications is less important than their overall diversity. Here the only focus is "score" -- that I interpreted as the number of classification errors found. You mention (eg line 842) you seek a "representative set", but my intuition is that given the objective, a RL system say on mnist dataset would directly "reward-hack" and only identify 8 misclassified for 0 (leaving all other confusions rather unexplored). Have you assessed the diversity/representativity along the score ?

-In my view, the experimental setting is unfair: RL auditor has already consumed "D_train" (=60% of complete test-set) examples. So concretely for example syntab, in figure 5, plots the range [3000,3050] of consumed requests... a fair baseline would have been to just submit these 3k examples and count the number of identified errors (as this constitutes the objective), which would represent according to table 1 (3k*0.14) around 400 errors: 12x more errors without additional complexity.

-I found the datasets choice rather exotic in a domain (classifiers). Why resort to home-brewed synthetic dataset when a plethora of classification data is readily available ?

**Questions:**

-Cf questions in the weaknesses part above.
-As I understand, your RL state space is continuous. This somehow deviates from the traditional RL (eg markovian) setting. Can you comment on the consequences of this choice ?

---

### Official Review · Reviewer_AejE · 2025-11-01

**Soundness:** 3
**Presentation:** 3
**Contribution:** 3
**Rating:** 6
**Confidence:** 3

**Summary:**

The paper frames auditing as an RL task and provides instantiations of various parts of an MDP to enable auditing. The paper conducts empirical experiments and user study to show the effectiveness of their framework.

**Strengths:**

(1) The idea of using RL for auditing is very interesting!

(2) The instantiations used in the paper make intuitive sense.

(3) The paper is well-written and thorough, in the sense that there are empirical experiments along with a user-study to show the efficacy.

**Weaknesses:**

(1) Set C is ill-defined. You are trying to construct C' which is similar to C, but what guarantees hold on C itself? I think this is a missing definition that is making the framework a bit on shaky grounds. Eg. C could be a set which consists of all the mispredicted inputs or the entire input space?

(2) why do humans still take 30 mins to complete auditing with your framework? is it the size of the dataset or the annotating process?

**Questions:**

see above

---

### Note · Authors · 2025-11-19

I have read and agree with the venue's withdrawal policy on behalf of myself and my co-authors.